# Prevalence of Co-Infections with Respiratory Viruses in Individuals Investigated for SARS-CoV-2 in Ontario, Canada

**DOI:** 10.3390/v13010130

**Published:** 2021-01-18

**Authors:** Adriana Peci, Vanessa Tran, Jennifer L. Guthrie, Ye Li, Paul Nelson, Kevin L. Schwartz, AliReza Eshaghi, Sarah A. Buchan, Jonathan B. Gubbay

**Affiliations:** 1Public Health Ontario, Toronto, ON M5G 1M1, Canada; Vanessa.Tran@oahpp.ca (V.T.); Jennifer.Guthrie@oahpp.ca (J.L.G.); Lennon.Li@oahpp.ca (Y.L.); Paul.Nelson@oahpp.ca (P.N.); Kevin.Schwartz@oahpp.ca (K.L.S.); alireza.eshaghi@oahpp.ca (A.E.); Sarah.Buchan@oahpp.ca (S.A.B.); 2University of Toronto, Toronto, ON M5S 1A1, Canada; 3Dalla Lana School of Public Health, University of Toronto, Toronto, ON M5T 3M7, Canada; 4Unity Health Toronto, Toronto, ON M5B 1W8, Canada; 5The Hospital for Sick Children, Toronto, ON M5G 1X8, Canada

**Keywords:** co-infection, SARS-CoV-2, COVID-19, seasonal respiratory viruses

## Abstract

Background: Co-infections of severe acute respiratory syndrome coronavirus 2 (SARS-CoV-2) with respiratory viruses, bacteria and fungi have been reported to cause a wide range of illness. Objectives: We assess the prevalence of co-infection of SARS-CoV-2 with seasonal respiratory viruses, document the respiratory viruses detected among individuals tested for SARS-CoV-2, and describe characteristics of individuals with respiratory virus co-infection detected. Methods: Specimens included in this study were submitted as part of routine clinical testing to Public Health Ontario Laboratory from individuals requiring testing for SARS-CoV-2 and/or seasonal respiratory viruses. Results: Co-infection was detected in a smaller proportion (2.5%) of individuals with laboratory confirmed SARS-CoV-2 than those with seasonal respiratory viruses (4.3%); this difference was not significant. Individuals with any respiratory virus co-infection were more likely to be younger than 65 years of age and male than those with single infection. Those with SARS-CoV-2 co-infection manifested mostly mild respiratory symptoms. Conclusions: Findings of this study may not support routine testing for seasonal respiratory viruses among all individuals tested for SARS-CoV-2, as they were rare during the study period nor associated with severe disease. However, testing for seasonal respiratory viruses should be performed in severely ill individuals, in which detection of other viruses may assist with patient management.

## 1. Introduction

Severe acute respiratory syndrome coronavirus 2 (SARS-CoV-2), the causative agent of coronavirus disease 2019 (COVID-19) was first identified in December 2019 in Wuhan, China, and has progressively spread, resulting in a global pandemic [1]. In Ontario, Canada, the first COVID-19 case was identified on 22 January 2020, with the number of daily cases peaking in the second week of April, during the first pandemic wave [2]. As of 22 May 2020, approximately 5 million cases and 300,000 deaths were reported worldwide, including more than 80,000 cases and almost 6000 deaths in Canada [3].

The disease is characterized by a wide range of clinical manifestations, from asymptomatic or mild symptoms (fever, cough, myalgia, and headache) to severe illness (pneumonia, acute respiratory distress, multiple organ failure) and death [4]. However, COVID-19 symptoms are non-specific to SARS-COV-2 as they are commonly reported with other respiratory pathogen infections [5,6,7,8,9,10].

Co-infections of SARS-CoV-2 with respiratory pathogens have been documented previously at varying rates. In a study of 1101 adult individuals with respiratory symptoms, in California, co-infection with another respiratory pathogen was reported in 24 (20.7%) of 116 persons with confirmed SARS-CoV-2 [6]. The most common secondary viruses identified were enterovirus/rhinonvirus and respiratory syncytial virus (RSV). Another retrospective study of hospitalized children in Wuhan reported that two (1.2%) of 161 children tested positive for co-infection of SARS-CoV-2 with additional viral and/or bacterial respiratory pathogens such as human metapneumovirus [hMPV] and RSV in one child and hMPV and *Mycoplasma pneumoniae* in the second child [7]. One of the children was severely ill, requiring intensive care unit (ICU) admission. Co-infections with bacteria and fungi, but not respiratory viruses, were reported among five of 99 severely ill patients in Wuhan [8].

Understanding the epidemiology and prevalence of seasonal respiratory viruses in patients with COVID-19 will help document the rate of SARS-CoV-2 co-infection and better appreciate the role of such viruses in patient’s clinical presentation. This could improve patient management and further contribute to public health practices aimed at virus containment measures.

The objectives of this study were to: assess the prevalence of co-infection of SARS-CoV-2 with seasonal respiratory viruses in various clinical settings, document the respiratory viruses detected among individuals tested for SARS-CoV-2, as well as describe characteristics of individuals with co-infection. We also describe and compare characteristics of individuals tested at Ontario’s public health laboratory (Public Health Ontario [PHO] Laboratory) for (i) both SARS-CoV-2 and seasonal respiratory viruses (SARS-CoV-2 + MRVP (multiplex respiratory virus polymerase chain reaction)) and (ii) seasonal respiratory viruses (MRVP) alone.

## 2. Methods

This study used a cross-sectional design. A total of 19,646 specimens were included in this study, submitted as part of routine clinical testing to PHO Laboratory from individuals seen in various hospitals, clinics, and assessment centers across the province. Specimens were tested for (i) SARS-CoV-2 and seasonal respiratory viruses (7225 specimens from 5228 individuals) or (ii) seasonal respiratory viruses alone (12,421 specimens from 11,542 individuals), based on tests requested on the laboratory requisition by the health care provider as well as the testing algorithm at PHO Laboratory [11]. Specifically, testing for SARS-CoV-2 was performed mostly for travel-related cases presenting to emergency rooms (ER) at the beginning of the pandemic, moving later towards broader criteria, including outbreaks in institutions. Testing for seasonal respiratory viruses alone was offered mostly to inpatients, institutionalized persons and those affected by respiratory outbreaks. Testing for seasonal respiratory viruses is not usually performed for patients seen in ambulatory/outpatient settings, or those seen in ERs, although it is provided on special request. The distribution of specimens by testing method is illustrated in Appendix A.

Testing and clinical information was extracted from the laboratory information management system (LIMS) at PHO Laboratory for the period 11 January 2020 to 20 April 2020. As the pandemic progressed, tests used and associated PHO Laboratory testing algorithms evolved to better address the increased needs and improve turnaround times.

During the study period, testing for SARS-CoV-2 was performed using three different methods: (i) a laboratory-developed endpoint nested polymerase chain reaction (PCR) assay targeting the RNA dependent RNA polymerase (RdRP) gene, followed by Sanger sequencing of amplicons with expected size of approximately 192 base pairs. This assay was adapted from a previously published Middle East Respiratory Syndrome Coronavirus (MERS-CoV) hemi-nested PCR, but altered such that the relevant primer bases match SARS-CoV-2: an outer primer and newly designed inner primers spanning 192bp were used for both amplification [12]; (ii) a laboratory developed real-time reverse-transcription (rRT)-PCR for specific detection of the SARS-CoV-2 envelope (E) gene and RdRp gene [13];(iii) the Roche cobas^®^ SARS-CoV-2 rRT-PCR assay on the cobas^®^ 8800 system, which detects the E gene and open reading frame (orf)1a/b gene. Initial specimens were confirmed by the National Microbiology Laboratory (NML) of the Public Health Agency of Canada using a nucleocapsid (N) gene rRT-PCR developed at NML. NML also conducted laboratory-developed conventional RT-PCRs targeting RdRp and ORF3a, followed by nucleotide analysis of partial gene sequences of RdRp and ORF3a amplicons. Detection of a single gene by any of the assays was considered positive for SARS-CoV-2. All tests were assessed for cross-reactivity with other respiratory viruses during validation and no-cross reaction was identified.

Testing for seasonal respiratory viruses at PHO Laboratory was performed using a laboratory-developed multiplex respiratory virus PCR assay (MRVP), which detects nine respiratory viruses including: adenovirus, seasonal human coronavirus (229E, NL63, OC43, HKU1), enterovirus, hMPV, parainfluenza (1–4), RSV A/B, rhinovirus, influenza A, influenza A(H3N2), influenza A(pdm09), and influenza B. MRVP was conducted using a laboratory developed three well real-time reverse transcription (rRT)-PCR assay developed from previously published protocols ([14,15]; additional references available on request). The assay includes the following targets: well 1: influenza A, H3 subtyping, influenza B, RSV, well 2: parainfluenza, enterovirus, H1pdm09 subtype, adenovirus, well 3: seasonal coronaviruses, rhinovirus, hMPV. The assay includes individual primers for parainfluenza 1–4, with a common probe, facilitating detection of all four parainfluenza viruses, but does not differentiate between them. Similarly, the assay detects the four human seasonal coronaviruses (OC43, NL63, HKU1 and 229E), without differentiating them. Each well also includes an internal control targeting human RNase to ensure adequate specimen quality, and control for nucleic acid extraction and amplification.

From 11 January–1 March 2020, all respiratory specimens submitted for SARS-CoV-2 testing were also routinely tested for other respiratory viruses. From 2 March–20 April 2020, testing for non-SARS-CoV-2 respiratory viruses was mainly conducted for inpatient or institutionalized (e.g., long-term care residents, correctional facility inmates) individuals when ordered on the laboratory requisition. Testing was also done for other patients by special arrangement. 

Data analyses were performed using Stata SE/10.0. Most analyses were performed at specimen level in order to retain individual’s characteristics (patient setting, clinical symptoms and geography) at the time of testing, particularly for those tested multiple times. Some patients were tested more than once for various different reasons such as diagnostic confirmation, virus clearance, contact tracing etc. Duplicate specimens were removed and data were analyzed at the specimen and individual level. Transformation to individual level was conducted only for the key findings.

Descriptive analyses were performed to characterize and compare specimens tested by (i) SARS-CoV-2 + MRVP with specimens tested by (ii) MRVP alone, with respect to age, gender, patient setting, Ontario health region, outbreak status, specimen type and reported symptoms. Proportion differences between these groups were compared using the chi-square test; a *p*-value of <0.05 was considered significant. The two groups were also compared with respect to the number of respiratory viruses identified.

Viruses identified in each specimen were documented and categorized as: co-infection, single infection or negatives. (i) Co-infection was defined as the presence of SARS-CoV-2 with at least one seasonal respiratory virus or presence of two or more seasonal respiratory viruses in the same specimen; (ii) a single infection was considered when only SARS-CoV-2 or seasonal respiratory virus was detected; (iii) and a negative result was defined as no detection of SARS-CoV-2 and/or seasonal respiratory virus.

The laboratory database was reviewed to identify study specimens that underwent additional molecular testing for the following pathogens: *Legionella* spp. *Bordetella pertussis*, *Mycoplasma pneumoniae* and fungi.

Adjusted logistic regression analyses were performed for specimens tested by (i) SARS-CoV-2 + MRVP to compare individuals with the likelihood of co-infection versus single infection (the outcome) adjusted for age group, gender, region, patient setting, outbreak related, group of viruses identified (SARS-CoV-2 or seasonal respiratory virus), and specimen type (exposure variables). Odds ratios (ORs) with 95% confidence intervals (CIs) were reported and interpreted.

### Ethics

The PHO Ethics Review Board has determined that this project did not require research ethics committee approval, as it describes analyses that were completed at PHO Laboratory as part of routine clinical respiratory testing during the first wave of the COVID-19 pandemic in Ontario and are, therefore, considered public health practice and exempt from this requirement.

## 3. Results

A mean of 1.5 specimens, (median 1 specimen; range 1 to 8 specimens) were tested per person of the group tested for SARS-CoV-2 and seasonal respiratory viruses. Among individuals with more than one specimen submitted, mean lag time between the first and any subsequent specimen tested was 0.2 days (median 0 days; range 0 to 32 days).

An average of 1.2 specimens (median 1 specimen; range of 1 to 6 specimens) were tested per person on the group tested for seasonal respiratory virus alone; the mean lag time to subsequent specimen tested was 13 days (median 0 days; range 0 to 86 days). 

Individuals tested by (i) SARS-CoV2 + MRVP versus (ii) MRVP alone differed for all study variables (Table 1). Persons tested for both SARS-CoV-2 and seasonal respiratory viruses were more likely to be younger (median age 65 versus 70 years), female (56.1% versus 51.6%), tested in an institution (38.8% versus 22.4%) and residing in Toronto (28.5% versus 19.5%) compared to persons tested for seasonal respiratory viruses alone. Additionally, such individuals were less likely to be tested as part of an outbreak investigation (26.3% versus 78.9%).

Of the 7225 specimens tested by (i) SARS-CoV2+MRVP, 2210 (30.6%) were positive for at least one respiratory virus compared to 4152 (33.4%) in the group tested by (ii) MRVP alone (*n* = 12,421) (*p* < 0.001). Of the specimens tested by (i) SARS-CoV-2 + MRVP, human seasonal coronaviruses 488 (6.8%) were the most common viruses detected, followed by SARS-CoV-2325 (4.5%) and rhinovirus 325 (4.5%) (Table 2). Five specimens with (i) SARS-CoV-2 + MRVP testing were also tested for *Legionella* spp. and were found to be negative. No other bacterial or fungal testing occurred in our cohort. Of specimens tested by (ii) MRVP alone, influenza A 1166 (9.4%) was the most common virus identified followed by human seasonal coronavirus 766 (6.2%) and RSV 564 (4.5%). Of influenza A specimens (*n* = 293) detected in the group tested by (i) SARS-CoV-2 + MRVP, 231 (78.8%) were influenza A/H1N1pdm09 and 46 (15.7%) as influenza A/H3N2. Influenza subtype distribution was 971 (83.3%) influenza A/H1N1pdm09 and 153 (13.1%) influenza A/H3N2 of all influenza A identified (*n* = 1166) in the group tested by (ii) MRVP alone. A small proportion of influenza A specimens (5.5% and 3.6% for each group, respectively) were not subtyped.

Total number of viruses identified were 2294 viruses and 4360 viruses in specimens tested by (i) SARS-CoV-2 + MRVP and (ii) MRVP alone, respectively. Of 2294 viruses identified in (i) SARS-CoV2 + MRVP group, 2129 were detected as single virus infection and 165 as co-infections of dual (*n* = 156) or triple (*n* = 9) viruses. Similarly, of 4360 viruses detected in the (ii) MRVP group, 3948 were detected as single virus infection and 412 as co-infection of dual (*n* = 400) or triple (*n* = 12) viruses.

Compared to specimens tested by (ii) MRVP (*n* = 12,421) alone, those tested by (i) SARS-CoV-2 + MRVP (*n* = 7225) had fewer viruses detected, whether as a single infection (2129 (29.5%) versus 3948 (31.8%)) or co-infection (81 (1.1%) versus 204 (1.6%)) (*p* < 0.001) (Table 3).

Co-infection of SARS-CoV-2 with a seasonal respiratory virus was detected in 8/325 (2.5%) of SARS-CoV-2-positive specimens and co-infection of seasonal respiratory viruses was identified in 81/1893 (4.3%) of specimens with seasonal respiratory viruses detected (*p* value > 0.05). Of SARS-CoV-2 co-infections, two had seasonal coronavirus, two had rhinovirus, two had RSV, and two had hMPV present. Of seasonal respiratory virus co-infections detected in the specimens tested by (i) SARS-CoV-2 + MRVP, influenza A/H1N1pdm09 and rhinovirus (*n* = 8) or adenovirus with seasonal coronavirus (*n* = 5) were the most common co-infections, while influenza A/H1N1pdm09 and seasonal coronavirus (*n* = 24) or RSV with seasonal coronavirus (*n* = 21) in the group tested by (ii) MRVP alone.

Characteristics of the eight patients with co-infection involving SARS-CoV-2 and a seasonal respiratory virus are shown in Table 4. The age of persons with SARS-CoV-2 co-infection was between 50–91 years (median age 75 years) and six were male. Fever and cough were the most common symptoms reported. Clinical setting for these eight patients was ambulatory or an institution. Interestingly, co-infection was not present in the 17 specimens with SARS-CoV-2 detected that were collected from severely ill patients in ICUs.

Reported symptoms were also compared between individuals tested by (i) SARS-CoV-2 + MRVP and (ii) MRVP alone, in Figure 1. The three most common symptoms reported by both groups, respectively, were: cough (2963 (41%) versus 1552 (12.7%)), fever (2563 (35.5%) versus 3238 (26.5%)) and undefined respiratory symptoms (2284 (31.6%) versus 4981 (40.7%)). Compared to individuals tested by (ii) MRVP alone, individuals tested by (i) SARS-CoV-2 + MRVP had a significantly higher number of reports for the following individual symptoms: cough, fever, sore throat, shortness of breath, nasal congestion, pneumonia, diarrhea, and chest pain (*p* value < 0.01); and they had less undefined respiratory symptoms, malaise, chronic obstructive pulmonary disease (COPD) and asthma; (*p* value < 0.01). However, this should be interpreted with caution as symptoms were significantly less reported for the (ii) MRVP group compared to the (i) SARS-CoV-2+MRVP group (4020/12,421 (32.4%) versus 890/7225 (12.3%)), respectively.

In the adjusted logistic regression analyses (Table 5), persons <65 years of age had significantly higher odds of being diagnosed with viral co-infection compared to single infection (OR = 3.1 and 95% CI (1.5–6.2)); the odds of being diagnosed with co-infection were 60% higher for males in comparison to females (OR = 1.6; 95% CI (1.0–2.5)). The odds of having a SARS-CoV-2 co-infection with another seasonal respiratory virus were lower than the odds for co-infection between two seasonal respiratory viruses; however, this difference was not significant.

## 4. Discussion

In this study we describe testing for SARS-CoV-2 and/or seasonal respiratory viruses at PHO Laboratory, Ontario’s reference microbiology laboratory. Persons tested for both SARS-CoV-2 and seasonal respiratory viruses were slightly younger than patients tested for seasonal respiratory viruses alone. This is likely because most individuals tested for seasonal respiratory viruses alone were tested as part of provincial outbreak investigations, representing mostly older adults residing in retirement homes and long-term care facilities. Early in the pandemic, SARS-CoV-2 testing was not routinely undertaken in retirement homes and long term care facilities, since at this time no virus was circulating in such settings. Individuals tested for SARS-CoV-2 and seasonal respiratory viruses were predominantly females, which is likely driven by SARS-CoV-2 testing. This is similar to findings in a provincial report describing characteristics of SARS-CoV-2 cases in Ontario [2].

Individuals tested for SARS-CoV-2 and seasonal respiratory viruses were seen mainly in ER and institutions. This could represent testing patterns for SARS-CoV-2 at the time, targeting travel-related cases presenting to (ER) in the beginning of the pandemic, moving later to other clinical settings. Those who were tested for seasonal respiratory viruses alone mostly received care in hospital which also reflects the testing algorithm at PHO Laboratory, offering respiratory testing mostly to inpatients and individuals residing in institutions [11].

Individuals tested for SARS-CoV-2 and seasonal respiratory viruses at PHO Laboratory were most likely from the Toronto area in comparison to those tested for respiratory viruses alone being from Central East Ontario. This resembles the population for which PHO Laboratory serviced, with SARS-CoV-2 testing in Ontario moving from centralized testing at PHO Laboratory in Toronto, in the beginning of the pandemic, to more distributed testing across other provincial hospitals and private laboratories as the pandemic progressed. Furthermore, as other laboratories implemented SARS-CoV-2 testing, they shifted from forwarding specimens to PHO Laboratory for both SARS-CoV-2 and seasonal respiratory virus testing to ordering seasonal respiratory virus testing only.

While the same respiratory viruses were detected in both groups, percent positivity for at least one respiratory virus was 30.6% in the group tested for both SARS-CoV-2 and seasonal respiratory viruses and 33.4% in the group tested for respiratory viruses only. Seasonal coronavirus was the most common virus detected in the first group and influenza A in the second one. This could be reflective of more specimens being tested for seasonal respiratory viruses early before SARS-CoV-2 fully evolved, which corresponds to the peak of the influenza season in Ontario. These two viruses were also the most common circulating viruses in Ontario at that time [16].

Co-infection with two or more respiratory viruses was detected in 1.1% of specimens tested for both SARS-CoV-2 and seasonal respiratory viruses and in 1.6% of specimens tested for respiratory viruses alone, with the most common being influenza A(H1N1)pdm09/ rhinovirus and adenovirus/seasonal coronavirus in the first group, and influenza A(H1N1)pdm09/ seasonal coronavirus and respiratory syncytial virus/seasonal coronavirus in the second group. Interestingly, percent positivity and percent of co-infections in this study were much lower compared to previously reported data in a community-acquired respiratory viruses co-infection study among patients of the sentinel practices network (SPSN) in Ontario, Canada [17]. In this study, at least one respiratory virus was identified among 65.6% of individuals and co-infection in 15.3% of tested individuals [17]. Results were lower in our study for two main reasons: first, unlike the SPSN study, we did not have any clinical enrollment requirements for patients being included in our study; secondly our study period included only the first four months of the COVID-19 pandemic, limiting the ability to capture several seasonal viruses such as enterovirus and rhinovirus, which typically circulate in summer–fall. Even influenza virus was not fully captured in our study, as the influenza season had already peaked in Ontario when the COVID-19 pandemic started [16]. However, a decrease of influenza activity was reported following the onset of the CoVID-19 pandemic, likely due to mitigation strategies put in place to reduce the spread of SARS-CoV-2 virus [18]. All of these factors combined may have led to both an underestimation and reduction in seasonal respiratory infection identified in this study, including co-infection.

This study found co-infection of SARS-CoV-2 with seasonal respiratory virus in 2.5% of SARS-CoV-2 positive specimens. Several studies investigating SARS-CoV-2 co-infection have reported varying rates of co-infection [4,5,6,7,8,19,20,21,22,23,24,25,26]. Wang et al. followed 8274 close contacts of COVID-19 cases in a university hospital in Wuhan, China, and found co-infection with respiratory viruses in 5.8% of 2745 patients with laboratory confirmed SARS-CoV-2 [19]. Conversely, a retrospective study among 257 positive laboratory-confirmed SARS-CoV-2 cases screened patients during hospital admission for 39 respiratory pathogens and found co-infection in 243 (94.2%) of individuals with either respiratory viruses (31.5%), bacteria (91.8%) and fungi (23.3%) [20]. Although some of these co-infections were likely colonization, most of them were documented within 1–4 days of COVID-19 disease onset, and individuals with SARS-CoV-2 had the most severe disease. The main reasons for differences in reported co-infection rates between studies rely on the population being investigated, the study period, testing methods used to identify secondary pathogens, and spectrum of secondary pathogens targeted.

We found being younger than 65 years of age and male increased the risk of co-infection. Similarly, Zhu et al. reported higher rates of co-infection among the 15–64 year old age group than those 65+ and children <15 years of age, but no differences in co-infection between females and males. Of note, their results were not adjusted for all variables in the study [20]. In our study, co-infection was neither that common nor significantly different for those with confirmed SARS-CoV-2 (2.5%) and those with seasonal respiratory viruses (4.3%). A systematic review had similar findings—prevalence of COVID-19 co-infection with another respiratory virus was reported to be 3%, with RSV and influenza A most common [27]. Co-infection could depend on season and also on the pathogenic competition between viruses as the risk of testing positive for SARS-CoV-2 was previously reported to be 58% lower among influenza positive cases [28].

In our study, persons with SARS-CoV-2 co-infection mostly reported mild respiratory symptoms including fever, cough and undefined respiratory symptoms. Apart from evidence that two of these individuals were seen in ER, there was no clear indication of disease severity. Furthermore, co-infection was not present in SARS-CoV-2 confirmed specimens collected from severely ill patients in ICUs. Being tested as part of an outbreak investigation and receiving care in an institution would indicate that the other six patients were likely elderly and, therefore, at risk for more severe respiratory disease. However, disease severity cannot definitively be established. SARS-CoV-2 co-infection with bacteria or fungi rather than respiratory viruses are reported to be potentially lethal in ICU patients [26]. A higher risk of death among individuals with SARS-CoV-2 and influenza co-infection than those with SARS-CoV-2 alone was previously reported [28]. These findings highlight the importance of considering testing for other respiratory pathogens (bacteria, viruses and fungi), particularly in critically ill COVID-19 patients. To the best of our knowledge, testing for other bacterial and fungal respiratory pathogens was rarely done in our cohort. Considering that 69.3% of specimens tested negative for SARS-CoV-2 or seasonal respiratory virus, it is important to investigate other causes that may be part of the differential.

### Limitations

There are several limitations in this study. First, our study included individuals who received testing at PHO Laboratory and, therefore, is not representative of all individuals tested in Ontario. Second, not all specimens tested for SARS-CoV-2 underwent testing for seasonal respiratory viruses. This would have limited the detection of the full spectrum of seasonal respiratory viruses present and underestimated co-infection. Third, testing methods for SARS-CoV-2 virus changed over time to fit pandemic needs. Differences in sensitivity and specificity between tests may exist, which may have caused some missed detections of SARS-CoV-2 and consequently fewer co-infection detections. Fourth, as a reference microbiology laboratory, PHO Laboratory does not perform primary bacteriology on respiratory specimens except for molecular testing for *Legionella* species, *Mycoplasma pneumoniae*, and *Chlamydia pneumoniae*. In addition, testing for bacterial and fungal pathogens was not broadly requested; therefore, we could not adequately examine co-infection of SARS-CoV-2 with bacteria or fungi. Lastly, PHO Laboratory does not have access to patient care charts, and relies on clinical information provided on the laboratory requisition, which may have prevented us from fully exploring disease severity.

## 5. Conclusions

Co-infection was detected in a smaller proportion (2.5%) of individuals with laboratory confirmed SARS-CoV-2 infection than in individuals with seasonal respiratory viruses (4.3%); however this difference was not significant. Individuals with any respiratory virus co-infection compared to those with single respiratory virus were more likely to be younger than 65 years of age and male. Those with SARS-CoV-2 co-infection manifested mostly mild respiratory symptoms, such as fever and cough; however due to scarcity of co-infection found in this study, its clinical implications cannot be conclusively determined.

In summary, findings of this study may not support routine testing for seasonal respiratory viruses among all individuals tested for SARS-CoV-2, as they were not commonly found during the study period nor clearly associated with severe disease. However, testing for seasonal respiratory viruses should be performed in severely ill individuals, in which detection of other respiratory viruses may assist with patient management.

## Figures and Tables

**Figure 1 viruses-13-00130-f001:**
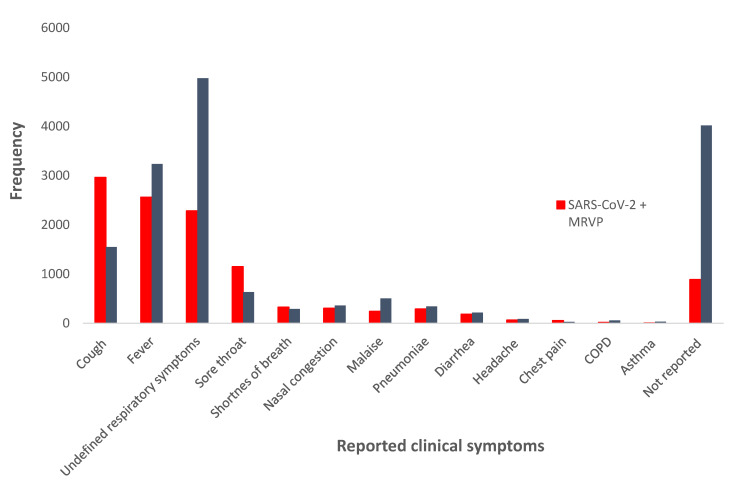
Comparison of reported symptoms among individuals tested by (i) SARS-CoV-2 + MRVP and (ii) MRVP. Counts in this figurerepresent the frequency of each symptom reported. More than one symptom was reported by many individuals; therefore the sum of symptoms frequency do not match with the total number of specimens tested. Undefined respiratory symptoms represent a general term recorded by clinicians on the requisition to indicate the presence of any respiratory symptoms without providing further specifics. Not all individuals had symptom information recorded in laboratory requisition. COPD—coronary obstructive pulmonary disease.

**Table 1 viruses-13-00130-t001:** Characteristics of patients tested for (i) severe acute respiratory syndrome coronavirus 2 (SARS-CoV-2) + MRVP (multiplex respiratory virus polymerase chain reaction) versus (ii) MRVP alone.

Variables/Categories	SARS-CoV-2 + MRVP (*n* = 7225)	MRVP (*n* = 12,421)	Chi-Square *p*-Value ^£^
**Median age in years (range)**	65 (0–107)	70 (0–107)	
**Age group**			<0.001
<5 years	230 (3.2)	908 (7.3)	
5–19 years	252 (3.5)	482 (3.9)	
20–49 years	1959 (27.1)	1816 (14.6)	
50–59 years	756 (10.5)	1140 (9.2)	
60–69 years	806 (11.2)	1654 (13.3)	
70–79 years	863 (11.9)	2146 (17.3)	
80–89 years	1369 (18.9)	2698 (21.7)	
90+ years	967 (13.4)	1537 (12.4)	
Unknown	23 (0.3)	40 (0.3)	
**Sex**			<0.001
Female	4053 (56.1)	6414 (51.6)	
Male	3166 (43.8)	5505 (44.3)	
Unknown	6 (0.1)	502 (4.1)	
**Patient setting**			<0.001
Ambulatory/No setting	1232 (17.1)	2711 (21.8)	
Emergency rooms (ER)	2343 (32.4)	390 (3.1)	
Hospitalization	633 (8.8)	5576 (44.9)	
Intensive-care unit (ICU)	213 (2.9)	964 (7.8)	
Institution	2804 (38.8)	2780 (22.4)	
**Ontario region**			<0.001
Central East	1833 (25.4)	3307 (26.6)	
Central West	1200 (16.6)	2418 (19.5)	
Eastern	668 (9.3)	1054 (8.5)	
North East	451 (6.2)	841 (6.8)	
North West	269 (3.7)	464 (3.7)	
South West	746 (10.3)	1911 (15.4)	
Toronto	2058 (28.5)	2426 (19.5)	
**Outbreak related**			<0.001
No	5325 (73.7)	2619 (21.1)	
Yes	1900 (26.3)	9802 (78.9)	
**Specimen type**			<0.001
Throat	1271 (17.6)	192 (1.6)	
NP	5665 (78.4)	11,212 (90.2)	
Other	289 (4.0)	1017 (8.2)	

Age was defined as age in years at time specimen was received at PHO Laboratory. Patient setting represented the location where patient had specimen collected. Institution represents retirement home, long term care facility, correctional facilities. Region was derived based on individual ‘postal code of residence or submitter’s postal code if individual’s postal code was not reported on laboratory requisition. Specimens were defined as outbreak- related when they were tested as part of an official public health unit declared outbreak or investigation and the remaining were considered non-outbreak related. NP, nasopharyngeal swab. Other specimen type category included nasal swab, broncho-alveolar lavage (BAL), eye, oral, swab undefined etc. ^£^
*p*-value is generated by comparing all categories of individual variables across the two testing groups.

**Table 2 viruses-13-00130-t002:** Respiratory viruses identified in specimens tested for (i) SARS-CoV-2 + MRVP versus (ii) MRVP alone.

Viruses Identified	SARS-CoV-2 + MRPV	MRPV
	(*n* = 7225)	(*n* = 12,421)
	Counts (%)	Counts (%)
SARS-CoV-2	325 (4.5)	0 (0.0)
Adenovirus	49 (0.7)	138 (1.1)
Seasonal coronavirus	488 (6.8)	766 (6.2)
Enterovirus	34 (0.5)	38 (0.3)
Rhinovirus	325 (4.5)	458 (3.7)
Influenza A	293 (4.1)	1166 (9.4)
Influenza B	182 (2.5)	439 (3.5)
hMPV	315 (4.4)	544 (4.4)
Parainfluenza	106 (1.4)	247 (2.0)
RSV	177 (2.4)	564 (4.5)

Counts represent frequencies of viruses identified in each group. Percent positivity is calculated based on total number of specimens tested in each group.

**Table 3 viruses-13-00130-t003:** Single infections and co-infections identified among specimens tested by (i) SARS-CoV-2 + MRVP and (ii) by MRVP alone.

Variable/Categories	SARS-CoV-2 + MRVP (*n* = 7225)	MRVP (*n* = 12,421)	Chi-Square *p*-Value *
Single infection	2129 (29.5)	3948 (31.8)	<0.001
Co-infection	81 (1.1)	204 (1.6)	
Negative	5015 (69.4)	8269 (66.6)	

Of 7225 specimens tested by (i) SARS-CoV-2+MRVP, SARS-CoV2 was identified in 325 specimens and seasonal respiratory viruses in 1893. Of 81 co-infections identified in this group, eight specimens (representing eight patients) had SARS-CoV-2 and a seasonal respiratory virus and 73 specimens had two or more seasonal respiratory viruses identified. * *p*-value is generated by comparing all infection categories (single infection, co-infection, negative) across the two testing groups.

**Table 4 viruses-13-00130-t004:** Characteristics of individuals co-infected with SARS-CoV-2 and a seasonal respiratory virus in individuals tested by (i) SARS-CoV-2 + MRVP.

								Reported Clinical Symptoms					
Patient	Specimen Collected Day	Specimen Type	Patient Setting	Age	Sex	SARS-CoV-2 Result	Seasonal Respiratory Virus Detected	Cough	Nasal Congestion	Fever	Undefined Respiratory Symptoms	Travel	Outbreak Related
1	01-Mar	NP	ER	50–59	M	Positive	Rhinovirus	Yes	NR	NR	NR	Yes	No
	01-Mar	Throat	ER	50–59	M	Positive	Negative	Yes	NR	NR	NR	Yes	No
	19-Mar	NP	Institution	50–59	M	Negative	Negative	Yes	NR	NR	NR	Yes	No
2	01-Mar	Throat	Institution	70–79	F	Positive	Negative	NR	Yes	Yes	NR	Yes	No
	01-Mar	NP	Institution	70–79	F	Positive	Rhinovirus	NR	Yes	Yes	NR	Yes	No
3	24-Mar	NP	Institution	80–89	M	Positive	hMPV	NR	NR	NR	NR	NR	Yes
4	27-Mar	NP	Institution	70–79	M	Positive	RSV A/B	NR	NR	Yes	Yes	NR	No
5	30-Mar	NP	Institution	90–99	M	Positive	Coronavirus	NR	NR	Yes	NR	NR	Yes
6	29-Mar	NP	Institution	80–89	F	Positive	hMPV	NR	NR	NR	Yes	NR	Yes
7	30-Mar	NP	ER	50–59	M	Positive	RSV A/B	Yes	NR	NR	NR	NR	No
8	31-Mar	NP	Institution	70–79	M	Positive	Coronavirus	Yes	NR	Yes	NR	NR	Yes

Mar—March; Institution represents retirement homes, long term care facilities, correctional facilities; undefined respiratory symptoms represents a general term recorded by clinicians on the requisition to indicate the presence of any respiratory symptoms without providing further specifics. NP, nasopharyngeal; ER, emergency room (not hospitalized); M, male; F, female; NR, feature was not mentioned on the laboratory requisition; RSV A/B—RSV A or B (MRVP does not distinguish between RSV types A and B).

**Table 5 viruses-13-00130-t005:** Crude and adjusted logistic regression comparing characteristics of individuals with respiratory virus co-infection versus single infection among those tested by (i) SARS-CoV-2 + MRVP.

Variables/Categorise	SARS-CoV-2 + MRVP (*n* = 2218)			
	Co-Infection (*n* = 89)	Single Infection (*n* = 2129)	Crude Results	Adjusted Results
	Counts (%)	Counts (%)	OR (95% CI)	OR (95% CI)
**Age group**				
65+	19 (21.3)	983 (42.3)	1	1
<65	70 (78.7)	1142 (53.7)	3.2 (1.9–5.3) *	3.1 (1.5–6.2) *
**Sex**				
F	35 (39.3)	1152 (54.1)	1	1
M	54 (60.7)	977 (45.9)	1.8 (1.2–2.8) *	1.6 (1.0–2.5) *
**Ontario region**				
Central	44 (49.4)	909 (42.7)	1	1
Eastern	8 (9.0)	171 (8.0)	1.0 (0.5–2.1)	1.2 (0.5–2.7)
South West	7 (7.9)	207 (9.7)	0.7 (0.3–1.6)	0.8 (0.3–1.7)
Toronto	28 (31.5)	678 (31.9)	0.9 (0.5–1.4)	0.8 (0.5–1.3)
North	2 (2.3)	164 (7.7)	0.3 (0.1–1.1)	0.3 (0.6–1.2)
**Outbreak-related**				
No	75 (84.3)	1514 (71.1)	1	1
Yes	14 (15.7)	615 (28.9)	0.5 (0.3–0.8) *	0.9 (0.4–2.2)
**Patient setting**				
Ambulatory/no setting	12 (13.5)	353 (16.6)	1	1
ER	51 (57.3)	826 (38.8)	1.8 (0.9–3.5)	1.4 (0.7–2.7)
Inpatient	9 (10.1)	192 (9.0)	1.4 (0.6–3.3)	1.7 (0.7–4.3)
Institution	17 (19.1	758 (35.6)	0.7 (0.3–1.4)	1.4 (0.5–3.8)
**Respiratory viruses**				
Seasonal respiratory viruses	81 (91.0)	1812 (85.1)	1	1
SARS-CoV-2	8 (9.0)	317 (14.9)	0.6 (0.3–1.2)	0.82 (0.4–1.8)
**Specimen type**				
Throat	26 (29.2)	423 (19.9)	1	1
Nasopharyngeal	62 (69.7)	1618 (76.0)	0.6 (0.4–1.0)	1.0 (0.6–1.7)
Other	1 (1.1)	88 (4.1)	0.2 (0.3–1.4)	0.2 (0.1–1.5)

This analysis was restricted to individuals with positive results for specimens tested by (i) SARS-CoV-2 + MRVP to allow investigation of SARS-CoV-2 co-infection. Co-infection category includes all specimens that had at least two respiratory viruses detected whether it was SARS-CoV-2 with a seasonal respiratory viruses or two or more seasonal respiratory viruses. Eight specimens with co-infection of SARS-CoV-2 were counted in both respiratory viruses categories because they include 8 SARS-CoV-2 viruses and 8 respiratory viruses. Single infection category includes all specimens that had only one respiratory virus. Some variables (age, region, patient setting) were aggregated further to reduce small sample size cells. Age was categorized as <65 and 65+. North Eastern and North Western regions were aggregated to North; and Central East and Central West regions were aggregated to Central. Persons receiving care in hospital or intensive care unit were aggregated to inpatients. No setting represents individuals for which setting was not recorded in the test requisition. Other specimen type category included nasal swab, bronchoalveolar lavage, eye, oral, swab undefined etc. * indicates *p*-value <0.05 and 95% CI do not cross 1.

## Data Availability

Public Health Ontario (PHO) cannot disclose the underlying data. Doing so would compromise individual privacy contrary to PHO’s ethical and legal obligations. Information about PHO’s data access request process is available on-line at https://www.publichealthontario.ca/en/data-and-analysis/using-data/data-requests.

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
