# Peer review of "Prevalence of Co-Infections with Respiratory Viruses in Individuals Investigated for SARS-CoV-2 in Ontario, Canada"

_viruses, 2021, doi:10.3390/v13010130_

Round 1
Reviewer 1 Report
The work of Peci et al. describes the prevalence of SARS-CoV-2 coinfections with other respiratory viruses in an impressive large Canadian cohort. The authors have many data but the presentation is rather insufficient.
It would help the reader it the authors provide a figure or table, which describes how many subjects were enrolled, tested with which test and were positive for which virus.
*) The Methods section is missing the description of the study cohort (how many) and what specimen were used. It would be interesting to include information why and how subjects were annotated to the two groups. Furthermore, it should be more structured regarding performed tests and much more detailed. E.g. there is no information how the multiplex test was performed.
*) The results are very unstructured and unclear to the reader. E.g line 157 says 4.5 % were infected with SARS-CoV-2 whereas figure 1 shows about 20 %. Indeed, figure 1 is very misleading and should be revised. As well as Figure 2.
*) Table 4 why was the adjusted logistic regression analyses performed?
*) Since Figure 2 shows symptoms of the SARS-CoV-2 group only as written in the figure legend, it would highly improve the manuscript if the authors provide a detailed description of symptoms and to compare the SARS-CoV-2+ MRVP with MRVP only to see if there are relevant differences.
*) Regarding the comparison of symptoms from line 204 on, the authors should not over interpret their data since the 8 subjects with SARS-CoV-2 and co-infections have a very incomplete symptoms data set. E.g Table 2 shows that no information was provided.
*) The authors should clarify their obtained results. According to the to date provided information the prevalence of coinfections with SARS-CoV-2 and respiratory viruses were rather low with no clinical implications observed regarding symptoms. This could also be an interesting result if presented accurate and clear.
Author Response
Reviewer 1
C1.1 The work Peci at al. describes the prevalence of SARS-CoV-2 co-infections with other respiratory viruses is an impressive large Canadian cohort. The authors have many data but the presentation is rather insufficient. It would help the reader if the authors provide a figure or table, which describes how many subjects were enrolled, tested with which test and were positive for which virus.
R1.1 We agree with reviewer’s comment and therefore we added a study diagram (Figure S1) in the supplemental materials. Figure S1 illustrates the distribution of specimens by testing method, results and number of viruses identified.
C1.2 The method section is missing the description of the study cohort (how many) and what specimens were used. It would be interesting to include information why and how subjects were annotated to the two groups. Furthermore it should be more structured regarding performed tests and much more detailed. E.g. there is no information how the multiplex test was performed.
R1.2a We agree with reviewer’s comment and therefore we added more details about the study cohort and a reference for the testing algorithm in the Methods section, lines 77-83 to read:
A total of 19,646 specimens were included in this study, submitted as part of routine clinical testing to PHO Laboratory from individuals seen in various hospitals, clinics, and assessment centers across the province. Specimens were tested for SARS-CoV-2 and seasonal respiratory viruses (7,225 specimens from 5,228 individuals) or seasonal respiratory viruses alone (12,421 specimens from 11,542 individuals), based on tests requested on the laboratory requisition by the health care provider as well as the testing algorithm at PHO Laboratory [11].
R.1.2b In the Methods section we also added another paragraph to describe how testing was distributed among the study population, at lines 83-39, to read:
Specifically, testing for SARS-CoV-2 was performed mostly for travel related cases presenting to emergency rooms (ER) at the beginning of pandemic, moving later towards broader criteria, including outbreaks in institutions. Testing for seasonal respiratory viruses alone was offered mostly to inpatients, institutionalized persons and those affected by respiratory outbreaks. Testing for seasonal respiratory viruses is not usually performed for patients seen in ambulatory/outpatient settings, or those seen in ER, though it is provided on special request. Distribution of specimens by testing method is illustrated in Figure S1.
C1.3 The results are very unstructured and unclear to the reader. E.g. line 157 says 4.5% were infected with SARS-CoV-2 whereas figure 1 shows bout 20%. Indeed figure 1 is very misleading and should be revised. As well as Figure 2.
R1.3 To make this easier we change the presentation of information in Figure 1 and replaced it with Table 2. Figure 2 was also changed to Figure 1 comparing symptoms between individuals tested between (i) SARS-CoV-2 + MRVP and (ii)MRVP alone as suggested in comment C1.5
C1.4 Table 4 why was the adjusted logistic regression analysis performed?
R1.4 One of the objectives of this manuscript was to describe characteristics of individuals with SARS-CoV-2 co-infection. Due to the small number of specimens identified with SARS-CoV-2 co-infection we were unable to investigate this thoroughly. However, by comparing co-infection (SARS-CoV-2 co-infection included) with single infection, adjusted for respiratory viruses (SARS-CoV2 and seasonal respiratory viruses) and other variables, we were able to tease out some characteristics of people with co-infection and more importantly compare the likelihood of co-infection between (i) SARS-CoV-2 and seasonal respiratory viruses versus (ii) co-infection between seasonal respiratory viruses alone. This finding is reported in the Discussion section at lines 276-278, and reads:
The odds of having a SARS-CoV-2 co-infection with another seasonal respiratory virus were lower than the odds for co-infection between two seasonal respiratory viruses; however, this difference was not significant.
C1.5 Since Figure 2 shows symptom SARS-CoV-2 group only as written in the figure legend, it would highly improve the manuscript if the authors provide a detailed description of symptoms and to compare the (i) SARS-CoV-2 + MRVP with (ii) MRVP only to see if there are relevant differences.
R1.5 We agree and therefore we changed the information in that figure (now Figure 1) by comparing reported symptoms of individuals tested by (i) SARS-CoV-2 + MRVP and (ii) MRVP alone. This information is illustrated in Figure 1. Results were also reported in the Results section, lines 251-261, to read:
Reported symptoms were also compared between individuals tested by (i) SARS-CoV-2 + MRVP and (ii) MRVP alone in Figure 1. The three most common symptoms reported by both groups, respectively, were: cough [2,963 (41%) versus 1,552 (12.7%)], fever [2,563 (35.5%) versus 3,238 (26.5%)] and undefined respiratory symptoms [2,284 (31.6% ) versus 4,981 (40.7%)]. Compared to individuals tested for (ii) MRVP alone, individuals tested by (i) SARS-CoV-2 + MRVP had a significantly higher number of reports for the following individual symptoms: cough, fever, sore throat, shortness of breath, nasal congestion, pneumonia, diarrhea, deaths and chest pain; they had less undefined respiratory symptoms, malaise, chronic obstructive pulmonary disease (COPD) and asthma; (p value <0.01 for all respective symptom comparisons). However, this should be interpreted with caution as symptoms were significantly less reported for the (ii)MRVP group compared to the(i) SARS-CoV-2+MRVP group [4,020/12,421 (32.4%) versus 890/7,225 (12.3%)], respectively.
C1.6 Regarding the comparison of symptoms from line 204 on the authors should not over interpret the data since the 8 subjects with SARS- CoV-2 and confections have a very incomplete symptoms data set. E. G Table 2 shows that no information was provided.
R1.6 We appreciate your comment. A slight change was made to table 4, NR replace NIR and we modified the relevant footnote: NR, feature was not mentioned on the laboratory requisition.
Clinicians use free text to describe symptoms manifested by their patient and they may not provide all the information when completing the requisition. This is raised in the Limitations section, lines 402-404:
Lastly, PHO Laboratory does not have access to patient care charts, and relies on clinical information provided on the laboratory requisition, which may have prevented us from fully exploring disease severity.
We are providing a link to our laboratory requisition for your further information.
https://www.publichealthontario.ca/-/media/documents/lab/2019-ncov-test-requisition.pdf?la=en%20https://www.publichealthontario.ca/-/media/documents/lab/2019-ncov-test-requisition.pdf?la=en
C1.7 The authors should clarify their obtained results. According to the data provided information the prevalence of co-infection with SARS-CoV-2 and respiratory viruses were rather low with no clinical implications observed regarding symptoms. This could also be an interesting result if presented accurate and clear.
R1.7 We agree and therefore we slightly modified our conclusions in lines 409-411 to read: Those with SARS-CoV-2 co-infection manifested mostly mild respiratory symptoms, such as fever and cough; however due to scarcity of co-infection found in this study, its clinical implications cannot be conclusively determined.

Reviewer 2 Report
This is an observational and cross-sectional epidemiological study. The study aims to establish the prevalence of coinfection of SARS-CoV-2 with seasonal respiratory viruses as compared to coinfections occurring in the case of seasonal respiratory viruses alone. The problem with this kind of study is a limited clinical utility due usually to a spate of uncontrolled, poorly controlled, or unmatched conditions. I'd like to mention just a few issues:
1/ What was actually the rationale to do it. One could expect that coinfection(s) could exacerbate the disease course, increase its severity or mortality, particularly in the case of Covid-19. I guess coinfections worsening the clinical course have been described, e.g., for influenza combined with RSV in neonates. Is it so in adults as well, any literature? Was it the background to collate these data or was there any other reason prompting you to do it? Anyhow, the study provides a negative answer as there were null differences in the prevalence of coinfections between Covid-19 and seasonal viruses (2.5%) and seasonal viruses, like influenza, versus other seasonal influenza-like respiratory viruses if I understand it well. Or the answer cannot be conclusively given due to a mismatch between the influenza season and the outbreak of Covid-19, differences in the sites of patient sampling, and many other factors.
2/ The first two paragraphs of results with the number of subjects and specimens per subject should be in the method section and not results. Actually, why you had more than one specimen (up to 6 or 8) taken from some same subjects. What was the reason for taking multiple samples one by one at very short time intervals, technical or clinical? What was the exact way of sampling and was it the same for all subjects?
3/ There were 3 types of PCR techniques used for the verification of Covid-19. Was it haphazard depending on the lab site or did it have any purpose? Would there be any difference between Covid-19 and the accompanying coinfections, depending on the PCR technique?
4/ To me the most interesting part would be coinfections with other pathogens like bacteria and fungi. To this end, the results are even less conclusive due to a meager occurrence of such cases or diagnostic difficulties.
5/ Another clinically important thing would be coinfections in severely ill Covid-19 patients. Again, no answer is possible since you strangely did not observe such patients.
6/ Finally, having said all that above and considering the neutral results, I must say I admire your diligent style of writing, perfect methodological and statistical approach to this study and the mention of quite a number of possible limitations. The article is professionally presented.
Author Response
Reviewer 2
This is an observational and cross-sectional epidemiological study. The study aims to establish the prevalence of co-infection of SARS-CoV-2 with seasonal respiratory viruses as compared to co-infection occurring in the case of seasonal respiratory viruses alone. The problem with this kind of study is a limited clinical utility due usually to a spate of uncontrolled, poorly controlled, or unmatched conditions. It would like to mention just a few issues:
C2.1 What was actually the rationale to do it? One could expect that co-infection(s) could exacerbate the disease course, increase its severity or mortality, particularly in the case of COVID-19. I guess co-infections worsening the clinical course have been described e.g. for influenza combined with RSV in neonates. Is it so in adults as well, any literature? Was it the background to collate these data or was there any other reason prompting you to do it? Anyhow the study provides a negative answer as there were null differences in the prevalence of co-infections between COVID-19 and seasonal viruses (2.5%) like influenza versus other seasonal influenza- like respiratory virus if I understand it well. Or the answer cannot be conclusively given due to a mismatch between the influenza season and the outbreak of COVID-19 differences in the sites of patient sampling and many other factors.
R2.1a We appreciate your comments and questions raised. It is true that cross-sectional studies have many limitations as you listed few of them above; however cross-sectional designs are quick and convenient and can help measure the prevalence of an event at one point in time. In our study we aimed to measure prevalence of SARS-CoV-2 co-infection with other respiratory pathogens and we believe it served us well.
R2.1b The emergence of a new virus (SARS-CoV-2) raised many questions that we needed to provide answers to, including the prevalence of co-infection with seasonal respiratory viruses being one of them. We wanted to know if this virus was able to co-circulate with other respiratory viruses? If so with which ones? How common was this phenomenon? What population was mostly affected? What were the clinical implications of SARS-CoV-2 co-infection? Do we need to routinely test patients for SARS-CoV-2 and seasonal respiratory -infections ? These and many more questions were the reasons that prompted us to undertake this study. We believe that the answers to these questions help clinicians and public health workers to better understand and manage these cases.
R2.1c In the Introduction and Discussion sections of the manuscript we have included findings from the existing literature to give more context to our work. The literature includes previous studies measuring SARS-CoV-2 co-infection as well as purely seasonal respiratory viruses co-infection among hospitalized individuals (adults and children), reporting co-infection of SARS-CoV-2 with respiratory viruses as well as respiratory bacteria, and also reporting the clinical implications of it. For example in the background section we have written the following paragraph, lines 54-64 which reads:
Co-infections of SARS-CoV-2 with respiratory pathogens have been documented previously at varying rates. In a study of 1,101 adult individuals with respiratory symptoms, in California, co-infection with another respiratory pathogen was reported in 24 (20.7%) of 116 persons with confirmed SARS-CoV-2 [6]. The most common secondary viruses identified were enterovirus/rhinonvirus and respiratory syncytial virus (RSV). Another retrospective study of hospitalized children in Wuhan reported that two (1.2%) of 161 children tested positive for co-infection of SARS-CoV-2 with additional viral and/or bacterial respiratory pathogens such as human metapneumovirus [hMPV] and RSV in one child and hMPV and Mycoplasma pneumoniae in the second child [7]. One of the children was severely ill, requiring intensive care unit (ICU) admission. Co-infections with bacteria and fungi, but not respiratory viruses, were reported among five of 99 severely ill patients in Wuhan [8].
R2.1e The uniqueness of our study is that we measure the prevalence of co-infection from various clinical settings including ambulatory, emergency department, inpatient, intensive care units and outbreaks. To illustrate this better we slightly modified the objectives in lines 79-71 to read:
The objectives of this study were to: assess the prevalence of co-infection of SARS-CoV-2 with seasonal respiratory viruses in various clinical settings, document the respiratory viruses detected among individuals tested for SARS-CoV-2, as well as describe characteristics of individuals with co-infection.
R2.1d With respect to conclusions, indeed we found a small proportion of SARS-CoV-2 co-infection with respiratory viruses among 2.5% of those with SARS-CoV-2 detected. We also found that the likelihood of SARS-CoV-2 with respiratory viruses is not significantly different from co-infection between two different respiratory viruses. That is also important to know as it may support the fact that this virus is not more likely than others to co-circulate. We also discussed and explained our low findings in the manuscript. Among other reasons, mitigation strategies put in place to reduce the spread of SARS-CoV-2 could play a role in decreasing other respiratory virus circulation and consequently causing low prevalence of SARS-CoV-2 and seasonal respiratory virus confection. We believe that for all these reasons the findings of this study are very important for health care providers and those who manage SARS-CoV-2 outbreaks.
C2.2 The first two paragraphs of results with the number of subjects and specimens per subject should be in the method section and not results. Actually, why you more than one specimen (up to 6 or 8 ) taken from some subjects ? What was the reason for taking multiple specimens one by one at very short time intervals, technical or clinical ? What was the exact way of sampling and was it the same for all subject ?
R2.2a These are all great questions and concerns. With regards to the data on number of subjects, we removed the information from the Results section and modified the Methods section to include the overall number of specimens as well as the total number of specimens for each group. We also included a study diagram (Figure S1) to better illustrate distribution of individuals/specimens in each group. The modified text in the Methods section, lines 77-83, is as follows:
A total of 19,646 specimens were included in this study, submitted as part of routine clinical testing to PHO Laboratory from individuals seen in various hospitals, clinics, and assessment centers across the province. Specimens were tested for (i) SARS-CoV-2 and seasonal respiratory viruses (7,225 specimens from 5,228 individuals) or (ii) seasonal respiratory viruses alone (12,421 specimens from 11,542 individuals), based on tests requested on the laboratory requisition by the health care provider as well as testing algorithm at PHO Laboratory [11].
R2.2b Regarding multiple testing: The population included in this study was tested as part of routine clinical testing. Individuals who presented with symptoms in various health care settings were swabbed and specimens sent to PHO Laboratory for testing. Clinicians may test a patient more than once for various different reasons including: diagnostic confirmation, monitoring for virus clearance, contact tracing etc. In addition, early in the pandemic, testing guidelines recommended submission of multiple specimens (e.g. throat and NP).
To make this clearer we included this sentence in the methods section, lines 133-135 to read: Some patients were tested more than once for various different reasons including diagnostic confirmation, virus clearance, contact tracing etc.
R2.2c Sampling methods: To inform and standardize specimen collection across the province, PHO Laboratory provided technical guidelines on its website about who to test and how to collect specimens for SARS-CoV-2 testing. However, PHO Laboratory neither can ensure compliance with these guidelines nor has control of exactly who gets tested.
C2.3 There were 3 types of PCR techniques used for the verification of COVID-19. Was it haphazard depending on the lab site or did it have any purpose? Would there be any difference between COVID-19 and the accompanying co-infections depending the PCR technique.
R2.3 This is another great comment. Indeed testing for SARS-CoV2 changed over the time during COVID-19 pandemic. In the beginning of pandemic our lab quickly designed an in- house PCR to be able to detect SARS-CoV-2 and respond to immediate pandemic needs. As the pandemic evolved, there were increased needs for testing with improved turnaround time. In response to this our lab designed PCR assays that could fulfill these requirements, without affecting test performance. In the study below by Gurtherie et al. conducted by our lab colleagues during the same period (submitted for publication), the overall PCR test sensitivity was reported to be 84.6%, with a negative predictive value of 95.5%. However, the possibility that we missed detection of SARS-CoV-2 and consequently miscount co-infections still exist. Therefore we have included the following text in the limitations section at lines 393-395
Third, testing methods for SARS-CoV-2 virus changed over time to fit pandemic needs. Differences in sensitivity and specificity between tests may exist, which may have caused some missed detections of SARS-CoV-2 and consequently fewer co-infection detections.
Jennifer L. Guthrie et al. Characteristics of SARS-CoV-2 Testing for Rapid Diagnosis of COVID-19 during the Initial Stages of a Global Pandemic. medRxiv 2020.12.23.20231589;
doi: https://doi.org/10.1101/2020.12.23.20231589
C2.4 To me the most interesting part would be co-infections with other pathogens like bacteria and fungi. To this end, the results are even less conclusive due to a meager occurrence of such cases of diagnostic difficulties.
R2.4 We agree. We were very interested to investigate the SARS-CoV-2 co-infection with other respiratory pathogens beyond respiratory viruses. For this reason, we screened all the specimens included in this study, for the most common respiratory pathogens that we test for in our lab, including Legionella species, Mycoplasma pneumoniae, Chlamydia penumoniae and fungi. We found few specimens that had test performed for this pathogens and we among these specimens there was no co-infection of SARS-CoV-2 with these respiratory pathogens. The main two reasons for this limitation are: a) PHO Laboratory is a reference laboratory and therefore we do not offer primary bacteriology testing on respiratory specimens; b) Our testing depends on what clinicians request for the patient on the requisition and these pathogens were not vastly requested. We have acknowledged both of these caveats in the limitation section, lines 397-401, which reads:
Fourth, as a reference microbiology laboratory, PHO Laboratory does not perform primary bacteriology on respiratory specimens except for testing for molecular testing for Legionella species, Mycoplasma pneumoniae, and Chlamydia pneumoniae. In addition, testing for bacterial and fungal pathogens were not broadly requested; therefore we could not adequately examine co-infection of SARS-CoV-2 with bacteria or fungi.
C2.5. Another clinically important thing would be co-infection in severely ill Covid-19 patients. Again, no answer is possible since you strangely did not observe such patients.
R2.5 We really appreciate this comment as addressing this would highlight an important fact about severity of SARS-CoV-2 co-infection that was previously overlooked. As mentioned above we did look at patients from various clinical settings including ICU, which represent severely ill patients. In the two groups investigated in this study, ICU patients contributed 213 (2.9%) specimens in the group tested by (i) SARS-CoV-2 + MRVP, and 964 (7.8%) specimens in the group tested by (ii) MRVP alone (Table 1). Among the first group, 17 specimens had SARS-CoV-2 identified and none of them had a second virus or bacteria identified. We thought it will be important to include this information in the Results and Discussion sections. The revised text appears at lines 239-241 (Results) lines 377-378 (Discussion) to read:
Lines 239- 241
Clinical setting for these eight patients was ambulatory or institution. Interestingly, co-infection was not present in the 17 specimens with SARS-CoV-2 detected that were collected from severely ill patients in ICU.
Lines 377-378
Further, co-infection was not present in SARS-CoV-2 confirmed specimens collected from severely ill patients in ICU.
C2.6 Finally, having said all that above and considering the neutral results, I must say I admire your diligent style of writing, perfect methodological and statistical approach to this study and the mention of quite a number of possible limitations. The article is professionally presented.
R2.6 Thank you!

Round 2
Reviewer 1 Report
One could see that the authors put effort to improve the manuscript.
I would like the authors to pay attention to perform changes in Figures or Table also in the plain text of the manuscript e.g on page 5 of the revised manuscript, Figure 1 in plain text refers to infection rates, although the authors changed this to Table 2.
Additionally, I would highly suggest to improve Figure 1 of the revised manuscript, by marking statistic significant differences. And remove "postmortem" as s symptom as well as "death" in the text. Death is not a symptom, the authors should delete or rephrase.
If this minor changes will be made, I see the manuscript suitable for publication.
Author Response
Response 2 to reviewer 1 comments:
C1.1 I would like the authors to pay attention to perform changes in Figures or Table also in the plain text of the manuscript e.g on page 5 of the revised manuscript, Figure 1 in plain text refers to infection rates, although the authors changed this to Table
R1.1 We appreciate reviewer’s comment and therefore Figure 1 in page 5 was changed to Table 2. We checked the entire document for any other miss referenced table or figure and found none.
C1.2 Additionally, I would highly suggest to improve Figure 1 of the revised manuscript, by marking statistic significant differences. And remove "postmortem" as s symptom as well as "death" in the text. Death is not a symptom, the authors should delete or rephrase.
R1.2 The Chi Square test were used to compare each symptom between the two groups and the p value was reported for all groups. We decided to improve this by adding specific p values for the higher and the lower reported symptoms, respectively.
Now the text reads: Reported symptoms were also compared between individuals tested by (i) SARS-CoV-2 + MRVP and (ii) MRVP alone, in Figure 1. The three most common symptoms reported by both groups, respectively, were: cough [2,963 (41%) versus 1,552 (12.7%)], fever [2,563 (35.5%) versus 3,238 (26.5%)] and undefined respiratory symptoms [2,284 (31.6% ) versus 4,981 (40.7%)]. Compared to individuals tested by (ii) MRVP alone, individuals tested by (i) SARS-CoV-2 + MRVP had a significantly higher number of reports for the following individual symptoms: cough, fever, sore throat, shortness of breath, nasal congestion, pneumonia, diarrhea, and chest pain (p value <0.01); and they had less undefined respiratory symptoms, malaise, chronic obstructive pulmonary disease (COPD) and asthma; (p value <0.01).
We agreed with the second comment and therefore we removed the “postmortem” from the list of symptoms illustrated in Figure 1.
